# Influence of Metal Oxide Particles on Bandgap of 1D Photocatalysts Based on SrTiO_3_/PAN Fibers

**DOI:** 10.3390/nano10091734

**Published:** 2020-09-01

**Authors:** Fail Sultanov, Chingis Daulbayev, Seitkhan Azat, Kairat Kuterbekov, Kenzhebatyr Bekmyrza, Baglan Bakbolat, Magdalena Bigaj, Zulkhair Mansurov

**Affiliations:** 1Faculty of Chemistry and Chemical Technology, al-Farabi Kazakh National University, Almaty 050000, Kazakhstan; sultanov.fail@kaznu.kz (F.S.); zmansurov@kaznu.kz (Z.M.); 2Laboratory of Energy-intensive and Nanomaterials, Institute of Combustion Problems, Almaty 050000, Kazakhstan; 3Institute of Chemical and Biological Technologies, Satbayev University, Almaty 050000, Kazakhstan; 4Faculty of Physics and Technology, L.N. Gumilev Eurasian National University, Nur-Sultan 010000, Kazakhstan; 5NanoBioMedical Centre, Adam Mickiewicz University, 61-614 Poznan, Poland; magdalena.bigaj@amu.edu.pl

**Keywords:** electrospinning, SrTiO_3_, fibers, photocatalytic, water splitting, bandgap, hydrogen

## Abstract

This paper deals with the study of the optical properties of one-dimensional SrTiO_3_/PAN-based photocatalysts with the addition of metal oxide particles and the determination of their bandgaps. One-dimensional photocatalysts were obtained by the electrospinning method. Particles of metals such as iron, chromium, and copper were used as additives that are capable of improving the fibers’ photocatalytic properties based on SrTiO_3_/PAN. The optimal ratios of the solutions for the electrospinning of fibers based on SrTiO_3_/PAN with the addition of metal oxide particles were determined. The transmission and reflection of composite photocatalysts with metal oxide particles were measured in a wide range of spectra, from the ultraviolet region (185 nm) to near-infrared radiation (3600 nm), to determine the values of their bandgaps. Thus, the introduction of metal oxide particles resulted in a decrease in the bandgaps of the obtained composite photocatalysts compared to the initial SrTiO_3_/PAN (3.57 eV), with the following values: −3.11 eV for SrTiO_3_/PAN/Fe_2_O_3_, −2.84 eV for SrTiO_3_/PAN/CuO, and −2.89 eV for SrTiO_3_/PAN/Cr_2_O_3_. The obtained composite photocatalysts were tested for the production of hydrogen by the splitting of water–methanol mixtures under UV irradiation, and the following rates of hydrogen evolution were determined: 344.67 µmol h^−1^ g^−1^ for SrTiO_3_/PAN/Fe_2_O_3_, 398.93 µmol h^−1^ g^−1^ for SrTiO_3_/PAN/Cr_2_O_3_, and 420.82 µmol h^−1^ g^−1^ for SrTiO_3_/PAN/CuO.

## 1. Introduction

Photocatalysis is a well-researched method for renewable energy production in the forms of solar energy and high-purity chemical fuel (H_2_, CH_4_/CH_2_OH) [1,2]. Photocatalytic water splitting occurs upon the solar light irradiation of a semiconductor photocatalyst and results in the formation of hydrogen (H_2_) and oxygen (O_2_) [3]. An advantage over conventional energy sources, like fossil fuels, is the lack of carbon monoxide production, which, in light of ongoing climate change debates, is a great benefit for the environment [4].

The material used in the photocatalytic system has to be chosen carefully. Strontium titanate (SrTiO_3_) is a wide-gap semiconductor that belongs to the perovskite family of ternary oxides with an ABO_3_ structure [5]. At room temperature, it exhibits a cubic structure with a lattice parameter a = 3.9053 Å [6]. Its attractive properties include strong catalytic activity, high chemical stability, and the long lifetime of electron-hole pairs [5]. Due to the bandgap energy of 3.2 eV, photo-excitation takes place with the use of light with a wavelength λ less than 387 nm (UV light) [2,7], which accounts for about 5% of solar energy [8]. The inherent gap edge positions can be modified by the implementation of elemental doping in the original semiconductor material [5,9]. Doping with metal or non-metal alloys can be applied to extend the activating spectrum, allowing the SrTiO_3_ photocatalyst to also be used in the visible light region [7,9]. Metal ions implemented in the semiconductor material become electron donors, enhancing the production of hydrogen [3,8,9]. Transition metal ions like Fe, Mn, Cu, Ni, and Cr have been shown to modify the bandgap position of semiconductor materials [10,11,12,13,14,15] without enhancing the formation of water, which is the case when noble metals (e.g., Rh, Pt) are employed [3,9]. The switch towards the visible light region was also achieved by doping titanate-based materials with Fe [9,16,17], Cr [5,18,19,20], and Cu [21,22].

The presence of metal in the semiconductor lattice alters electron-hole recombination. Since the transfer of the trapped electron and hole to the semiconductor surface is required for the photocatalytic reaction to occur, it is important that the metal ions are located near the surface of the semiconductor for a more efficient charge transfer [9]. In addition, the effective separation of charges is significantly affected by the specific surface area, which promotes the free diffusion of water [23], as well as the high degree of crystallinity of photocatalysts, leading to a decrease in the number of recombinations of photogenerated charges [24]. Moreover, for the successful use of photocatalysts for the production of hydrogen by water splitting, it is necessary to develop and create inexpensive, efficient, and stable photocatalytic systems, which can result in a decrease in the market price of hydrogen [25].

In our previous work, we reported a low-cost synthesis and thorough characterization of SrTiO_3_ nanofibers (up to 350 nm in diameter) using the electrospinning method [26]. Here, we show that the modification of this material using oxides of particles of iron (Fe), chromium (Cr), and copper (Cu) results in a narrowing of the bandgap energy. The used metal oxides have different properties for trapping and transferring electrons and holes. An increase in the efficiency of the hydrogen evolution reaction from the water-alcohol mixture under visible light irradiation (λ > 400 nm), without disrupting the nanofibers, structure, or crystallinity of SrTiO_3_, was observed for all of the metal oxide particles added to SrTiO_3_ nanofibers. The results of this work show that metal oxide particles added to SrTiO_3_ nanofibers create visible-light-responsive photocatalysts, making them potent candidates for the conversion of solar energy to fuel.

## 2. Materials and Methods

Strontium nitrate (Sr(NO_3_)_2_, 98%), titanium oxide (TiO_2_, 99%), and oxalic acid ((COOH)_2_·2H_2_O, 98%) were purchased from Laborpharma (Almaty, Kazakhstan). Polyacrylonitrile (PAN, average M.W. is 152000), dimethylformamide (99.8%), iron chloride (FeCl_3_, 45% solution), copper oxide (CuO, 99.995% powder), and chromium sulfate (Cr_2_(SO_4_)_3_, 99.99% powder) were purchased from Sigma Aldrich (St. Louis, MO, USA). All chemicals were used without further purification.

### 2.1. Electrospinning of SrTiO_3_/PAN-Based Fibers with the Addition of Metal Oxide Particles

SrTiO_3_ was obtained as described in [27]. A precursor for the electrospinning of fibers based on SrTiO_3_ and metal oxide particles was prepared as follows: PAN was used to create the polymer solution by its dissolution in dimethylformamide under constant stirring for 30 min. Then, SrTiO_3_ powder and FeCl_3_, Cr_2_(SO_4_)_3_, or CuO were added to the polymer solution at different ratios and stirred until the mixture became homogeneous. The obtained suspension was used as a precursor for obtaining fibers based on SrTiO_3_ with metal oxide particles by pulling under high voltage. Fiber electrospinning was carried out at room temperature with a voltage of 16 kV and a flow rate of 1.5 mL/h. The collector was located at a distance of 15 cm from the needle. Aluminum foil with a 20 cm diameter was used as a collector, which was replaced every 1.5 h throughout the entire process. The obtained fibers were then thermally stabilized at 185 °C for 15 min and calcined at 500 °C for 30 min in an argon medium. The resulting samples were designated as SrTiO_3_/PAN/Fe_2_O_3_, SrTiO_3_/PAN/CuO, and SrTiO_3_/PAN/Cr_2_O_3_ depending on the added metal.

### 2.2. X-ray Diffraction Analysis of Samples

The X-ray diffraction (XRD) analysis was carried out on a Dron-4-type X-ray diffractometer (Omsk, Russian Federation) with a range of rotation angles for diffraction unit detection from −100° to 168°. The minimum step for moving the detection unit is 0.001°. The permissible connection of the detection unit from a given rotation angle is ± 0.015°. The transport rate of the goniometer is 820°/min. The main error in measuring the pulse count of the X-ray measurement was not more than 0.4%.

### 2.3. Scanning Electron Microscope Characterization of the Surface Morphology of Samples

The surface morphology of the obtained photocatalytic fibers was studied using a Quanta 3D 200i (Waltham, MA, USA) scanning electron microscope (SEM) under an accelerating 15 kV voltage.

### 2.4. Measurement of the Transmission and Reflection of SrTiO_3_/PAN Fibers with the Addition of Metal Oxide Particles in a Wide Spectral Region from Ultraviolet (185 nm) to Near-Infrared Radiation (3600 nm)

The transmission and reflection measurements of photocatalytic fibers with the addition of metal oxide particles in a wide spectral region, from ultraviolet (185 nm) to near-infrared radiation (3600 nm), were carried out on a Shimadzu UV-3600 spectrophotometer (Moscow, Russian Federation) equipped with three detectors: a photoelectron multiplier for operation in the ultraviolet and visible spectral range, a semiconductor InGaAs, and cooled PbS detectors for near-infrared operation.

### 2.5. Investigation of the Activity of Photocatalysts Based on SrTiO_3_/PAN Fibers with the Addition of Metal Oxide Particles

The activity of photocatalysts based on SrTiO_3_/PAN fibers with the addition of metal oxide particles was tested by measuring the output of hydrogen during the water–methanol mixture splitting under UV radiation. The mixture, containing a photocatalyst, water, and methanol in different ratios, was loaded into a quartz tube reactor, which was previously purged with inert gas (argon) and exposed to UV irradiation with a wavelength of 320 nm and a power source of 40 W. As a result of the photocatalytic reaction of the water–methanol mixture splitting, the evolved mixture of gases was accumulated in a sealed sampler. The qualitative and quantitative composition of the evolved mixture of gases was analyzed by gas chromatography on a Chromos 1000 chromatograph (Dzershinsk, Russian Federation) with three packed 3 mm columns filled with NAX and PORAPAK Q phases, allowing for the identification of the leading gases: hydrogen, nitrogen, oxygen, carbon monoxide, and carbon dioxide.

## 3. Results and Discussion

### 3.1. The Synthesis of Fibers Based on SrTiO_3_/PAN with the Addition of Metal Oxide Particles and a Study of Their Physicochemical Properties

To achieve a highly efficient photocatalyst based on SrTiO_3_ fibers, it is necessary to create its composites with metal oxide particles. The addition of metal oxide particles allows for a narrowing of the bandgap of SrTiO_3_, leading to the possible use of a wide spectrum of visible light. It also contributes to the improvement of redox reactions that occur during the absorption of light. The experimentally selected optimal ratios of the solution components for obtaining fibers with the required characteristics are 1:9:2:88 SrTiO_3_:PAN:FeCl_3_:solvent, 0.5:10:1.5:88 SrTiO_3_:PAN:Cr_2_(SO_4_)_3_:solvent, and 1.5:8:2.5:88 SrTiO_3_:PAN:CuO:solvent.

Figure 1 presents SEM images of the obtained polymer fibers based on SrTiO_3_/PAN with metal oxide particles added. Polymer fibers based on SrTiO_3_/PAN with the addition of metal oxide particles have a continuous cylindrical shape without defects and are randomly arranged. The samples have the typical structure of fibers obtained by electrospinning, in which they are in contact with each other, forming a three-dimensional polymer network [28]. The average diameter of the obtained fibers is in the range from 200 to 400 nm, which is directly proportional to the viscosity of the solution used for electrospinning and the high voltage applied [29]. According to the SEM images, the presence of metal oxide particles and SrTiO_3_ does not affect the morphological characteristics of the forming polymer fibers, which is also confirmed by the results obtained in [30], in which the effect of the composition of the electrospinning solution on the diameters of such fibers was studied. For all types of obtained fibers, the size of the agglomerates of SrTiO_3_ and metal oxides ranges from 1 to 4 μm.

The inclusion of metal oxides does not cause a corresponding change in the fiber diameter. On the one hand, the metal particle addition increases the viscosity of the solution, but on the other hand, this effect is balanced by an increase in the electrical conductivity of the initial solution, which contributes to the formation of thinner fibers due to an increase in the charge density on the electrospinning jet, and this, in turn, leads to an elongation of the jet along its axis [31,32].

To confirm the presence of metal oxide particles in the calcined fiber structure based on SrTiO_3_/PAN, an XRD analysis of the samples was performed. Figure 2 presents the X-ray diffraction patterns of calcined fibers based on SrTiO_3_/PAN and their composites with metal oxide particles.

Figure 2a shows that for calcined SrTiO_3_/PAN-based fibers, characteristic peaks are observed at 22.77°, 32.41°, 39.99°, 46.49°, 57.81°, 67.86°, and 77.17°, indicating the presence of perovskite-type SrTiO_3_ in the structure. These peaks are also present in the X-ray diffraction patterns for samples of SrTiO_3_/PAN with the addition of metal oxide particles (Figure 2b), indicating the presence of SrTiO_3_ in each sample (JCPDS:35-0734). At the same time, for samples of SrTiO_3_/PAN with the addition of metal oxide particles, peaks with a lower intensity are also observed: for Fe_2_O_3_, the characteristic peaks are at 33.14°, 35.68°, 49.51°, 52.34°, and 54.07°; for Cr_2_O_3_, they are at 22.79°, 24.50°, 33.66°, 36.25°, 41.58°, 63.54°, and 65.24°; and for CuO, they are at 22.74°, 35.46°, 36.46°, 38.66°, and 52.32° (Figure 2b). According to the XRD analysis, for the SrTiO_3_/PAN/Fe_2_O_3_ sample, the content of SrTiO_3_ in the fibers is 89.6 wt.%, while the content of Fe_2_O_3_ is 10.4 wt.% (Figure 2b, black line). The crystal lattice parameter for SrTiO_3_ is 3.9036 Å (for the standard compound, it is 3.90010 Å), indicating a good crystallinity of the obtained samples of calcined fibers based on SrTiO_3_/PAN/Fe_2_O_3_. The crystallite size of the SrTiO_3_/PAN/Fe_2_O_3_ composite is 740 Å. Figure 2b (red line) presents the X-ray diffraction pattern of calcined fibers based on SrTiO_3_/PAN/Cr_2_O_3_; the content of SrTiO_3_ in the structure is 54.3 wt.%, while the content of Cr_2_O_3_ is 45.7 wt.%. The higher content of Cr_2_O_3_ compared to Fe_2_O_3_ is associated with the molar masses and densities of Cr_2_(SO_4_)_3_ and FeCl_3_, which were used as additives in the process of electrospinning the composite fibers. The molar volume of Cr_2_(SO_4_)_3_ is twice as high as that of FeCl_3_, which is confirmed by the semi-quantitative analysis of the calcined fibers based on SrTiO_3_/PAN/Cr_2_O_3_ and SrTiO_3_/PAN/Fe_2_O_3_. The crystallite size for fibers based on SrTiO_3_/PAN/Cr_2_O_3_ is 840 Å, and the crystal lattice parameter for SrTiO_3_ is 3.9036 Å. According to the results of the XRD analysis of the calcined fibers based on SrTiO_3_/PAN /CuO (Figure 2b, blue line), the content of SrTiO_3_ is determined to be 77.8 wt.%, and the content of CuO is 21.2 wt.% The crystallite size for the fiber based on SrTiO_3_/PAN/CuO is 760 Å. 

### 3.2. Investigation of the Transmission and Reflection Spectra of the Obtained Photocatalytic Fibers 

The photocatalysis mechanism of SrTiO_3_ is based on the formation of electron–hole pairs under UV irradiation, where they have sufficiently high energy for the formation of radicals with a high oxidation ability. To study the possible use of a wider spectrum, including visible light, for SrTiO_3_/PAN-based photocatalysts with the addition of metal oxide particles, their transmission and reflection spectra were determined. Analysis of the transmission spectra allows for the calculation of the bandgap of SrTiO_3_-based photocatalysts with the addition of metal oxides. For crystalline semiconductors, the following equation is valid for the relationship between the absorption coefficient and the incident photon’s energy: *α*(*ϑ*)*hϑ* = B (*hϑ* − *E*_gap_)*^m^*(1)
where *E_gap_* is the optical bandgap, B is a constant, *hϑ* is incident photon energy, and α(υ) is an absorption coefficient, which is in accordance with the law of Beer–Lambert, equal to
*α*(*ϑ*) = 2303*Ab*(*λ*)/*d*(2)
where *d* is film thickness and *Ab(λ)* is the film absorption coefficient.

For a more accurate determination of *α*, a correction that accounts for the reflection spectrum for the absorption coefficient must be made. To calculate the bandgap of SrTiO_3_/PAN-based photocatalysts with the addition of metal oxide particles, it is necessary to rewrite Equation (1):*α*(*ϑ*) = *B*(*hc*)^(*m* − 1)^*λ*(1/*λ* − 1/*λ_g_*)*^m^*(3)
where *λ_g_* is the wavelength corresponding to the bandgap, *h* is the Planck constant, and *c* is the speed of light.

Using the Beer–Lambert law, Equation (3) can be rewritten as follows:*Ab*(*λ*) = *B*(*hc*)^(*m* − 1)^*d*/2303(1/*λ* − 1/*λ_g_*)*^m^* + *B*_1_(4)
where *B*_1_ is the constant that takes into account the reflection spectrum.

Using Equation (4), the optical bandgap can be calculated by fitting the absorption spectrum without considering the film thickness. To determine the bandgap (*E*_gap_), the absorption coefficient’s dependence on the incident radiation energy was plotted and a linear approximation was carried out. Figure 3 presents the absorption and reflection spectra (Figure 3a) and a graph of the absorption coefficient’s dependence on the incident radiation energy for SrTiO_3_/PAN-based photocatalysts with the addition of metal oxide particles (Figure 3b).

As a result, the bandgap for SrTiO_3_/PAN-based photocatalysts with Cr_2_O_3_ particles added is determined to be 2.89 eV (Figure 3b, red line). Thus, it is found that the addition of Cr_2_O particles to SrTiO_3_-based fibers narrows the bandgap to 2.89 eV, making it possible to use a wide radiation spectrum. The determined value of the bandgap for a SrTiO_3_/PAN-based photocatalyst with Cr_2_O_3_ particles added can be explained by the occupied level of Cr^3+^ cations, which is 1.0 eV above the valence band. The bandgap of the SrTiO_3_/PAN-based photocatalyst with CuO particles added is 2.84 eV (Figure 3b, blue line). The determined bandgap of the SrTiO_3_/PAN-based photocatalyst with the addition of Fe_2_O_3_ particles is 3.11 eV (Figure 3b, black line). A high bandgap value for a photocatalyst with Fe_2_O_3_ particles added is associated with a low Fe_2_O_3_ content in calcined fibers, confirmed by XRD (Figure 1b, black line). In turn, the calculated bandgap of the photocatalyst based on the initial SrTiO_3_/PAN fibers without metal oxide particles added is 3.57 eV.

### 3.3. Investigation of the Activity of Photocatalysts Based on SrTiO_3_/PAN Fibers with the Addition of Metal Oxide Particles by the Output of Hydrogen during the Splitting of Water–Methanol Mixture

After determining the bandgaps of photocatalysts based on SrTiO_3_/PAN fibers with the addition of metal oxide particles, their photocatalytic efficiencies in the splitting of water–methanol mixtures with the production of hydrogen were studied. As seen in Table 1, the composition of the photocatalyst significantly affects the efficiency of hydrogen evolution. As reported in previous work [26], the photocatalytic hydrogen evolution rate from the splitting of the water-methanol mixture using SrTiO_3_/PAN-based fibers at a 40W UV radiation is 305.96 μmol/h. In turn, the results of the measurement of the average rate of hydrogen evolution during the splitting of the water–methanol mixture at 400W UV irradiation of the composite SrTiO_3_/PAN-based fibers with the addition of metal oxides are the following: 344.67 μmol/h for the SrTiO_3_/PAN/Fe_2_O_3_ photocatalyst, 398.93 μmol/h for the SrTiO_3_/PAN/Cr_2_O_3_ photocatalyst, and 420.82 μmol/h for the SrTiO_3_/PAN/CuO photocatalyst. The higher rates of hydrogen evolution for photocatalysts based on SrTiO_3_/PAN fibers with metal oxides added are explained by the fact that the optical, thermal, and electrochemical properties of metal oxide particles, which also highly depend on their sizes, allow not only for a narrowing of the bandgap of the semiconductors used as a photocatalyst but also for an improvement of their ultraviolet absorption ability. 

A comparison of the photocatalytic activity of different photocatalysts for hydrogen production shows that the composition of the photocatalyst and the type and intensity of irradiation significantly influence the intensity of hydrogen evolution. The rate of photocatalytic hydrogen evolution from a water–organic alcohol mixture using SrTiO_3_/PAN-based fibers with the addition of metal oxide particles during UV radiation with 40 W power is several times higher than that of reference analogs [32,33,34,35,36]. Moreover, irradiation with a power of 300 to 400 W was used in these works, which is not economically profitable, in contrast to lamps with a power of 40 W. In [34], TiO_2_ doped with platinum particles was used as a photocatalyst. The power of ultraviolet radiation was 125 W, and the rate of photocatalytic hydrogen evolution using this photocatalyst with platinum was 523.71 μmol/h, which is still comparable to the hydrogen evolution rate for the SrTiO_3_/PAN/CuO-based photocatalyst, while the cost and applied irradiation power required to conduct photocatalysis are much lower. 

## 4. Conclusions

One-dimensional photocatalysts based on SrTiO_3_/PAN fibers with the addition of metal oxide particles were obtained by the electrospinning method. A study of the transmission and reflection spectra showed that the addition of Cr_2_O_3_, CuO, and Fe_2_O_3_ particles led to decreases in the bandgaps of SrTiO_3_/PAN-based photocatalysts to 2.89, 2.84, and 3.11 eV, respectively. As a result of the addition of metal oxide particles to the initial SrTiO_3_/PAN-based photocatalyst, the rate of hydrogen evolution in the photocatalytic splitting of a water–methanol mixture increased to 344.67 µmol h^−1^ g^−1^ for the photocatalyst based on SrTiO_3_/PAN/Fe_2_O_3_, 398.93 µmol h^−1^ g^−1^ for the photocatalyst based on SrTiO_3_/PAN/Cr_2_O_3_, and 420.82 µmol h^−1^ g^−1^ for the photocatalyst based on SrTiO_3_/PAN/CuO. We believe that this article’s proposed approach for increasing the efficiency of photocatalysts by the addition of non-expensive materials, allowing for a reduction in their bandgap, is a promising method for the further development of technologies for efficient solar hydrogen production. 

## Figures and Tables

**Figure 1 nanomaterials-10-01734-f001:**
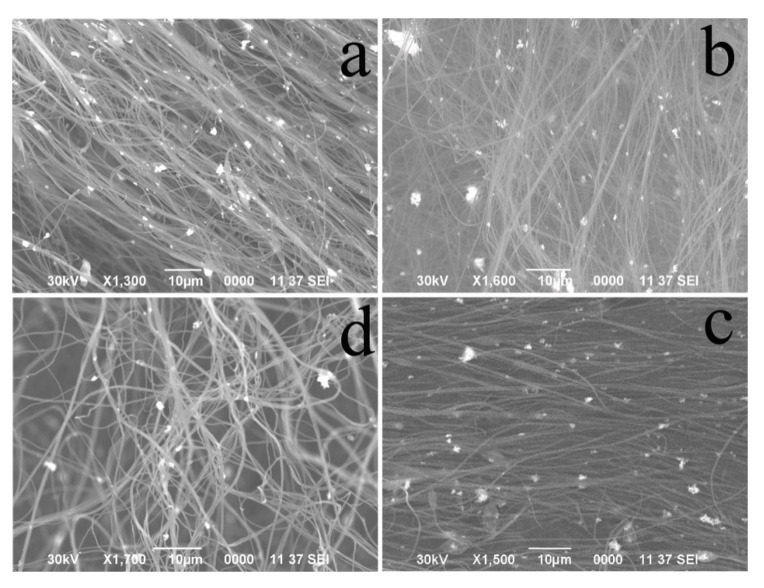
SEM images of non-calcined polymer fibers based on SrTiO_3_/PAN (**a**) and their composites with metal oxide particles: SrTiO_3_/PAN/Fe_2_O_3_ (**b**), SrTiO_3_/PAN/Cr_2_O_3_ (**c**), and SrTiO_3_/PAN/CuO (**d**).

**Figure 2 nanomaterials-10-01734-f002:**
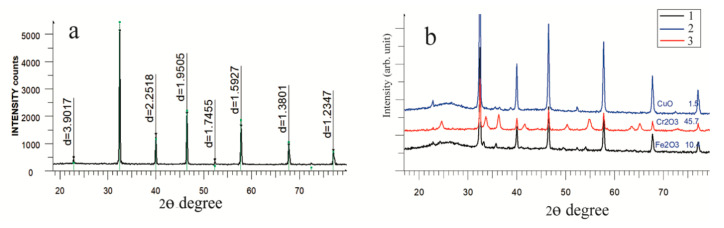
X-ray diffraction patterns of calcined fibers based on SrTiO_3_/PAN (**a**) and SrTiO_3_/PAN with the addition of metal oxide particles (**b**): SrTiO_3_/PAN/Fe_2_O_3_—1 (black line); SrTiO_3_/PAN/CuO—2 (blue line); SrTiO_3_/PAN/Cr_2_O_3_—3 (red line).

**Figure 3 nanomaterials-10-01734-f003:**
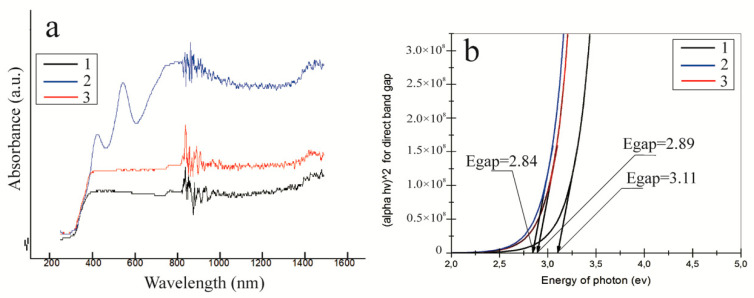
Absorption and reflection spectra (**a**) and values of the bandgap (**b**) of synthesized composite photocatalysts: SrTiO_3_/PAN/Fe_2_O_3_—1 (black line); SrTiO_3_/PAN/CuO—2 (blue line); and SrTiO_3_/PAN/Cr_2_O_3_—3 (red line).

**Table 1 nanomaterials-10-01734-t001:** Comparison of the photocatalytic activity of different photocatalysts for producing hydrogen by splitting mixtures of water and alcohol.

Type of Photocatalyst	Parameters of the Process	Composition of the Water Mixture	The Output of Hydrogen, µmol h^−1^ g^−1^	Reference
SrTiO_3_/PAN-based fibers	40 W UV lamp, quartz reactor	80% of water and 20% of CH_3_OH	305.96	[26]
SrTiO_3_/PAN/Fe_2_O_3_ fibers	40 W UV lamp, quartz reactor	80% of water and 20% of CH_3_OH	344.67	This work
SrTiO_3_/PAN/Cr_2_O_3_ fibers	40 W UV lamp, quartz reactor	80% of water and 20% of CH_3_OH	398.93	This work
SrTiO_3_/PAN/CuO fibers	40 W UV lamp, quartz reactor	80% of water and 20% of CH_3_OH	420.82	This work
MoSe_2_/TiO_2_	Xe arc lamp	90% of water and 10% of CH_3_OH	4.9	[33]
SrTiO_3_ doped with Cr and N	300 W xenon lamp, quartz reactor	81.5% of water and 18.5% of CH_3_OH	106.7	[34]
Pt/TiO_2_	125 W xenon lamp, quartz reactor	70% of water and 30% of CH_3_OH	523.71	[35]
Pt/ZrO_2_/TaO*_n_*	300 W mercury lamp, quartz reactor	85% of water and 15% of CH_3_OH	9	[36]

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
