# Peer review of "Influence of Metal Oxide Particles on Bandgap of 1D Photocatalysts Based on SrTiO_3_/PAN Fibers"

_nanomaterials, 2020, doi:10.3390/nano10091734_

Round 1

Reviewer 1 Report

My questions / comments are given below:

1)      Authors are recommended to add more quantitative information in the abstract.

2)      Introduction part needs to be expended with some articles. Some of the published articles may be considered:

  • Onwubiko, I., Khan, W.S., Subeshan, B., and Asmatulu, R. “Investigating the Effects of Carbon-Based Counter Electrode Layers on the Efficiency of Hole-Transporter-Free Perovskite Solar Cells,” Energy, Ecology and Environment, Vol. 5, pp. 141-152, 2020.
  • Moniruddin, M., Ilyassov, B., Zhao, X., Smith, E., Serikov, T., Ibrayev, N., Asmatulu, R., and Nuraje, N. “Recent Progress on Perovskite Nanomaterials in Photovoltaic and Water Splitting Applications,” Materials Today Energy, Vol. 7, pp. 246-259, 2018.
  • Alharbi, A., Alarifi, I. M., Khan, W.S., Swindle, A., and Asmatulu, R. “Synthesis and Characterization of Electrospun Polyacrylonitrile/Graphene Nanofibers Embedded with SrTiO3/NiO Nanoparticles for Water Splitting,” Journal of Nanoscience and Nanotechnology, Vol. 17, pp. 1-9, 2017.
  • Asmatulu, R., Shinde, M. A., Alharbi, A., and Alarifi, I. M. “Integrating Graphene and C60 into TiO2 Nanofibers via Electrospinning Process for the Enhanced Energy Conversion Efficiencies,” Macromolecular Symposia, Vol. 365, pp. 128-139, 2016.
  • Lie, Y., Asmatulu, R., and Nuraje, N. “Photo-Active Metal Oxide Nanomaterials for Water Splitting,” ScienceJet, Vol. 4, pp. 169-173, 2015.

3)      English and grammar needs to be checked for further improvement.

4)      Overall, the manuscript is good, and can be considered for a publication in your journal.

Author Response

Responses to Reviewer 1.

Dear Reviewer, thanks a lot for your review, which indicates the weak parts of the article, this is really helpful to increase the its quality. The article was fully revised in accordance with the following comments:

  • Authors are recommended to add more quantitative information in the abstract.

Response of authors:

The abstract was added by more quantitative information on investigated photocatalysts. See the lines 26-30 of the revised article.

  • Introduction part needs to be expended with some articles. Some of the published articles may be considered:

Response of authors:

The introduction part was extended with using some of the indicated published articles, see the lines 58-65 of the revised article.

  • English and grammar needs to be checked for further improvement.

Response of authors:

English and grammar was checked and revised throughout the article.

Reviewer 2 Report

The authors present an interesting work on the enhancement of the photocatalytic properties of SrTiO3 nanowires by the addition of various metal particles to increase the photoabsortion.

While the paper is of interest to researchers in the field of photocatalysis there are some issues that need addressing before it is complete.

1) the experimental section line 74 lists "salts" as one of the starting materials but doesn't specify which salt.

2) the method implies that Fe, Cu and Cr are added to all samples when this is not the case.

3) table 1 needs a better description. I assume this table shows the different ratios that were tested for synthesis but doesn't say what each achieved and why they were chosen and how the method used for each sample was chosen as only three samples are discussed throughout the rest of the manuscript.

4) line 128 - I am not sure why this statement appears here. it is summarised in the introduction and if it is to stay in the results it should appear at the start.

5) Figure 1 has an error in the caption. image (a) is mentioned twice and (b) is not mentioned. I assume that the second (a) should be (b).

6) All of the SEM images are of different magnifications. it would be useful if they were all of the same magnification for ease of comparison for th reader.

7) There is no discussion of the control SrTiO3/PAN sample. what are the agglomerates that can be seen in this image?

8) In the sentence starting in line 145

"In case of the fibers based on SrTiO3/PAN with addition of chromium and copper particles, the density of the forming polymer network is higher than that of fibers based on SrTiO3/PAN and fibers based on SrTiO3/PAN with addition of iron particles (Figure 1 c, d)."

this sentence is very confusing because of the positioning of the figure ref as it implies that the figures referenced at the end are the Fe samples.

9) 151 - the authors state that the diameters are directly proportional to the viscosity of the precursors. do the authors have any evidence for this, was viscosity measured and recorded as a control of fibre diameter? Or can they add a reference to support this statement?

10) similar to 9. Line 155 do they have any evidence for the increased conductivity of the samples (was it measured?) and its effect'?

11) Lines 158-178 this (while it is useful information) seems oddly placed and feels more like it should be in the introduction

12) Figure 2. The XRD patterns need a full characterisation. all of the peaks need to be tabled and assign. also there is not pattern for the metal free control sample.

13) the discussion starting at line 185 does not mention which sample is being discussed. 

14) In section 3.2 can the authors provide the data for the transmission and reflectance data for all of the samples along with the plots for band gap calculation. 

15) figure 4 why is there no data for Cr sample?

Author Response

Responses to Reviewer 2.

Dear Reviewer, thanks a lot for your review, which indicates the weak parts of our article, this is helpful to improve its quality. The article was fully revised in accordance with the following comments:

1) the experimental section line 74 lists "salts" as one of the starting materials but doesn't specify which salt.

Response of authors:

The experimental part was revised carefully and with indicating the types of used materials (salts and oxides), see the line 85 of the revised article.

2) the method implies that Fe, Cu and Cr are added to all samples when this is not the case.

Response of authors:

FeCl3, Cr2(SO4)3, and CuO were added separately to SrTiO3/PAN electrospinning solution to obtain the 3 types of photocatalysts: SrTiO3/PAN/Fe2O3, SrTiO3/PAN/CuO, and SrTiO3/PAN/Cr2O3 depending on the added metal. This part was improved, see the 2.1 Electrospinning of SrTiO3/PAN based fibers with additions of metal oxide particles, lines 81-93 of the revised article.

3) table 1 needs a better description. I assume this table shows the different ratios that were tested for synthesis but doesn't say what each achieved and why they were chosen and how the method used for each sample was chosen as only three samples are discussed throughout the rest of the manuscript.

Response of authors:

Table 1 was deleted since she does not carry any valuable information. We have left only the optimal parameters for electrospinning: “The experimentally selected optimal ratios of the solution components for obtaining fibers with the required characteristics are: SrTiO3:PAN:FeCl3:solvent - 1:9:2:88, SrTiO3:PAN:Cr2(SO4)3:solvent - 0.5:10:1.5:88 and SrTiO3:PAN:CuO:solvent - 1.5:8:2.5:88”. See the lines 132-135 of the revised article.

4) line 128 - I am not sure why this statement appears here. it is summarised in the introduction and if it is to stay in the results it should appear at the start.

Response of authors:

Thanks for your comment. This part was deleted, see the revised article.

5) Figure 1 has an error in the caption. Image (a) is mentioned twice and (b) is not mentioned. I assume that the second (a) should be (b).

Response of authors:

Thank you for pointing this out. SEM image 1 b was changed taking into account the remark, and their description was also edited (lines 136-146).

6) All of the SEM images are of different magnifications. it would be useful if they were all of the same magnification for ease of comparison for the reader.

Response of authors:

Thanks. We have attached SEM images of similar magnifications; see the Figure 1 of revised article.

7) There is no discussion of the control SrTiO3/PAN sample. what are the agglomerates that can be seen in this image?

Response of authors:

The discussion of SEM images was revised, please see the line 136-146 of the revised article.

8) In the sentence starting in line 145

"In case of the fibers based on SrTiO3/PAN with addition of chromium and copper particles, the density of the forming polymer network is higher than that of fibers based on SrTiO3/PAN and fibers based on SrTiO3/PAN with addition of iron particles (Figure 1 c, d)."

this sentence is very confusing because of the positioning of the figure ref as it implies that the figures referenced at the end are the Fe samples.

Response of authors:

We are agree. The submitted draft of article had serious problems with Figure captures, references and their description. The description of SEM images of obtained photocatalysts was revised, please see the lines 136-146 of the revised version of article.

9) 151 - the authors state that the diameters are directly proportional to the viscosity of the precursors. do the authors have any evidence for this, was viscosity measured and recorded as a control of fibre diameter? Or can they add a reference to support this statement?

Response of authors:

The references to support this statement are added (line 155).

10) similar to 9. Line 155 do they have any evidence for the increased conductivity of the samples (was it measured?) and its effect'?

Response of authors:

The references to support this statement are added (line 155).

11) Lines 158-178 this (while it is useful information) seems oddly placed and feels more like it should be in the introduction

Response of authors:

Thanks for the comment. The indicated part of the text was deleted

12) Figure 2. The XRD patterns need a full characterisation. all of the peaks need to be tabled and assign. also there is not pattern for the metal free control sample.

Response of authors:

The description of the XRD has been updated with revised detailed description (line 163-185). XRD for the sample of photocatalyst based on SrTiO3/PAN without addition of metal oxide prticles was included (figure 2a).

13) the discussion starting at line 185 does not mention which sample is being discussed.

Response of authors:

The description of results of XRD was improved. See the lines 163-185 of the revised version of article

14) In section 3.2 can the authors provide the data for the transmission and reflectance data for all of the samples along with the plots for band gap calculation.

Response of authors:

Absorption and reflection spectra and values of bandgap of investigated materials were included (Figure 3 a and b).

15) figure 4 why is there no data for Cr sample?

Response of authors:

In revised version of article the data on absorption and reflection spectra and values of bandgap for all 3 types of metal oxide added photocatalysts (SrTiO3/PAN/Fe2O3, SrTiO3/PAN/CuO, SrTiO3/PAN/Cr2O3) are added, see the revised Figure 3 and the description (lines218-228).

Reviewer 3 Report

The Authors report on the influence of metal particles on bandgap of SrTiO3/PAN fibers.  

Numerous papers have already demonstrated the effect of metal particles and ions on semiconductors properties. Therefore, the novelty contained in this work is poor.

However, I recommend major revisions before acceptance for publication.

  • As described in the section 2.1. Formation of SrTiO3/PAN fibers, metal doped SrTiO3/PAN fibers are properly produced by electrospinning process and then calcined at 500°C.

In the section Results and Discussion it is not clear if the different characterizations are carried out on calcined or not calcined fibers.

It is quite obvious that SEM investigations were carried out on not calcined materials (in fact, PAN decomposes at about 300°C), whereas XRD analyses on calcinated ones. However, this must be clearly reported in the text.

  • Figure 1 shows SEM images of SrTiO3/PAN fibers and SrTiO3/PAN fibers doped by Fe, Cr and Cu. Figures 1 a, c and d have scale bar of 10 micron. Only Figure 1 b displays a scale bar of 5 micron. Because there is not reason to use different scales, the Authors must replace Figure 1b.
  • In order to evaluate metals distribution on the SrTiO3/PAN fibers SEM-EDS investigations must be performed.
  • At pages 5-6, lines 190-196, the Authors report “The molecular chains of the polymer, SrTiO3 and metal particles are not bounded with each other and exist in the form of separate polymer helices and particles in solution. If the polymer concentration in the electrospinning solution increases, then its chains penetrate into each other more often, this leading to further restoration of the solvent, by evaporation, before the polymer chains transfer to crystallites. Generally, the process is presented by melt crystallization which is also known as bulk polymerization”. This sentence can be considered a hypothesis that needs more data to be verified.

Author Response

Responses to Reviewer 3.

Dear Reviewer, thanks a lot for your review, which indicates the weak parts of our article, this is helpful to improve its quality. The article was fully revised in accordance with the following comments:

1) As described in the section 2.1. Formation of SrTiO3/PAN fibers, metal doped SrTiO3/PAN fibers are properly produced by electrospinning process and then calcined at 500°C.

In the section Results and Discussion it is not clear if the different characterizations are carried out on calcined or not calcined fibers.

It is quite obvious that SEM investigations were carried out on not calcined materials (in fact, PAN decomposes at about 300°C), whereas XRD analyses on calcinated ones. However, this must be clearly reported in the text.

Response of authors:

Thanks for the comment. In the revised article, we tried to make the article more understandable for reader and indicate what materials are characterized. SEM images are captured as: “SEM images of non-calcined polymer fibers based on SrTiO3/PAN (a) and its composites with metal oxide particles: SrTiO3/PAN/Fe2O3 (b), SrTiO3/PAN/Cr2O3 (c) and SrTiO3/PAN/CuO (d)”. The capture of Figure 2 (XRD analysis) is the following: “Figure 2 - X-ray diffraction patterns of calcined fibers based on SrTiO3/PAN (a) and SrTiO3/PAN with additions of metal oxide particles (b): SrTiO3/PAN/Fe2O3 – 1 (black line); SrTiO3/PAN/CuO – 2 (blue line); SrTiO3/PAN/Cr2O3 – 3 (red line)”. Please see the revised version of the article.

2) Figure 1 shows SEM images of SrTiO3/PAN fibers and SrTiO3/PAN fibers doped by Fe, Cr and Cu. Figures 1 a, c and d have scale bar of 10 micron. Only Figure 1 b displays a scale bar of 5 micron. Because there is not reason to use different scales, the Authors must replace Figure 1b.

Response of authors:

Thanks. We have attached SEM images of similar magnifications; see the Figure 1 of revised article.

3) In order to evaluate metals distribution on the SrTiO3/PAN fibers SEM-EDS investigations must be performed.

Response of authors:

Yes, you are right, but unfortunately, due to circumstances and lack of time (7 days for revision) we are not able to conduct EDS-analysis. At least, article contains the data of XRD analysis to confirm the presence of SrTiO3 and metal oxides in the synthesized composite photocatalysts.

4) At pages 5-6, lines 190-196, the Authors report “The molecular chains of the polymer, SrTiO3 and metal particles are not bounded with each other and exist in the form of separate polymer helices and particles in solution. If the polymer concentration in the electrospinning solution increases, then its chains penetrate into each other more often, this leading to further restoration of the solvent, by evaporation, before the polymer chains transfer to crystallites. Generally, the process is presented by melt crystallization which is also known as bulk polymerization”. This sentence can be considered a hypothesis that needs more data to be verified.

Response of authors:

This unnecessary information was deleted from the article

Round 2

Reviewer 2 Report

I appreciate the changes that the authors have made to the manuscript. I would now recommend it for publication after one minor question.

In figure 3 the absorption spectra for the Fe2O3 sample shows a series of absorbance bands do this correspond to other bandgaps?

Author Response

Dear Reviewer, thanks for your valuable comments on our article, they are very useful to improve its quality.

We appreciate that you are recommending the revised version of the article for publication in the “Nanomaterials”.

Reviewer’s comment:

In figure 3 the absorption spectra for the Fe2O3 sample shows a series of absorbance bands do this correspond to other bandgaps?

Author’s response:

All the calculation of the bandgap were conducted in accordance with the equations (1-4), using the data of the absorption and reflection spectra of synthesized materials. The absorbance bands of the absorption spectra for SrTiO3/PAN/Fe2O3 composite photocatalyst (Figure 3) does not correspond to other bandgaps, so the value of bandgap of this composite photocatalyst was calculated as 3.11 eV.

Reviewer 3 Report

In my opinion, after the corrections/modifications carried out by the authors the manuscript can be accepted for publication.

Author Response

Dear Reviewer, thanks for your valuable comments on our article, they are very useful to improve its quality.

We appreciate that you are recommending the revised version of the article for publication in the “Nanomaterials”.
